# Apple Leaf Disease Identification with a Small and Imbalanced Dataset Based on Lightweight Convolutional Networks

**DOI:** 10.3390/s22010173

**Published:** 2021-12-28

**Authors:** Lili Li, Shujuan Zhang, Bin Wang

**Affiliations:** 1College of Agricultural Engineering, Shanxi Agricultural University, Jinzhong 030800, China; lilycqdxyjs@163.com (L.L.); wangbin1759@126.com (B.W.); 2College of Information Science and Engineering, Shanxi Agricultural University, Jinzhong 030800, China

**Keywords:** RegNet, apple leaf disease identification, complex environment, imbalanced dataset

## Abstract

The intelligent identification and classification of plant diseases is an important research objective in agriculture. In this study, in order to realize the rapid and accurate identification of apple leaf disease, a new lightweight convolutional neural network RegNet was proposed. A series of comparative experiments had been conducted based on 2141 images of 5 apple leaf diseases (rust, scab, ring rot, panonychus ulmi, and healthy leaves) in the field environment. To assess the effectiveness of the RegNet model, a series of comparison experiments were conducted with state-of-the-art convolutional neural networks (CNN) such as ShuffleNet, EfficientNet-B0, MobileNetV3, and Vision Transformer. The results show that RegNet-Adam with a learning rate of 0.0001 obtained an average accuracy of 99.8% on the validation set and an overall accuracy of 99.23% on the test set, outperforming all other pre-trained models. In other words, the proposed method based on transfer learning established in this research can realize the rapid and accurate identification of apple leaf disease.

## 1. Introduction

China is the largest fruit-producing country in the world, with the total area planted and the total yield of apples ranking first in the world. However, the apple leaf is easily affected by diseases and pests, such as ring rot, rust, early defoliation, scab, and so on. In most cases, it is not only subjective but also time-consuming, laborious, and inefficient to rely on agricultural and forestry experts for field identification or farmers to identify apple leaf diseases according to their experience. Farmers with less experience are more likely to misjudge and use pesticides blindly, which not only can fail to prevent the diseases, but can also affect the quality and yield of apples and cause environmental pollution, thus causing unnecessary economic losses. Therefore, it is of great significance to realize intelligent, fast, and accurate apple leaf disease identification.

In recent years, with the development of machine learning technology and the enhancement of computer data processing capabilities, researchers use image processing, machine learning, and other methods to automatically identify crop diseases, such as those in rice [1,2] corn [3], wheat [4], cotton [5], tomato [6] and cucumber [7,8]. By using color, shape, texture, and other information to establish feature vectors, and then using an artificial neural network (ANN) [9], support vector machine (SVM) [10] or other methods to classify feature vectors, a certain classification effect can be achieved. For example, Furferi et al. [11] clearly demonstrated the feasibility of a hybrid machine vision-ANN-based approach for the predictive purposes of an olive ripening index. However, traditional image processing methods need to extract disease features through manual design according to the feature of the disease, which has some problems, such as unstable feature extraction and poor adaptability. In particular, the influence of plant type, growth stage, disease type, environmental aspects, and other factors leads to the difficulty of feature extraction.

With the development of deep learning technology, convolutional neural networks, which can realize end-to-end detection by learning features in different fields, different scenes, and different scales, has become the research hotspot of the automatic identification of plant diseases and insect pests in agriculture.

The author Zhou et al. [12] proposed a disease identification model based on “image-text” multimodal collaborative representation and knowledge assistance (ITK-Net) and achieved an identification accuracy of 99.63% on a dataset of tomato and cucumber diseases. Wang et al. [13] proposed a two-stage model that fuses DeepLabV3+ and U-Net for cucumber leaf disease severity classification (DUNet) in complex backgrounds and reached an average accuracy of 93.27% and 92.85% for leaf segmentation and disease severity classification, respectively. Li et al. [14] used the VGGNet-16 and Inception V3 models to classify the different degrees of ginkgo leaf disease. Results showed that the VGGNet-16 model achieved an average accuracy of 98.44% and 92.19% under laboratory conditions and field conditions, respectively, while the Inception V3 achieved average accuracies of 92.3% and 93.2%, respectively.

Although the deep network model performs well in the task of plant leaf disease recognition, considering that some application scenarios need to be transplanted to embedded or mobile devices with limited hardware resources, this model still faces many constraints, such as a large number of model parameters, long training time and need for a large storage space [15,16]. Therefore, the present research tends to the miniaturization and practicability of the model structure. Sun et al. [17] proposed a variety of improved AlexNet models by using batch normalization, dilated convolution, and global pooling, which reduced the parameters of the model and improved the recognition accuracy. Guo et al. [18] proposed a multiscale receptive field recognition model for a mobile platform based on AlexNet, in which different sizes of convolution kernels were set for the first layer of the model, and various features were extracted to characterize the dynamic changes of the disease. Liu et al. [19] proposed two kinds of lightweight convolutional neural network models based on MobileNet and InceptionV3 for plant disease classification, which considered the recognition precision, operation speed, and model size, and realized plant leaf disease detection in the mobile phone terminal.

Instead of starting the training from scratch by randomly initializing weights, transfer learning initializes the weights using a pre-trained network on large labeled datasets, such as the public image datasets ImageNet, etc. Thus, the transfer learning method can reduce the amount of data needed in the training process and shorten the time spent in the training process, allowing a fewer number of images to be utilized to yield highly accurate trained networks [20].

Considering that the apple leaf disease dataset is a small dataset, the pre-trained lightweight network RegNet [21] was applied to the classification of apple leaf disease images with complex backgrounds for the first time and compared with the state-of-the-art models MobileNetV3 [22], ShuffleNet [23], EfficientNet-B0 [24], and Vision Transformer (ViT) [25].

The remainder of this paper is organized in the following manner. Section 2 introduces the collection of the image dataset and discusses the methodology to accomplish the task of apple leaf disease identification, along with related concepts of the proposed approach. Section 3 presents the arrangement of experiments, and multiple experiments are conducted. The experimental results are evaluated and a comparative analysis is performed, which is discussed in Section 4. Finally, this paper is concluded in Section 5.

## 2. Materials and Methods

### 2.1. Image Dataset

#### 2.1.1. Image Dataset Acquisition

The diseased apple leaf images were collected using smartphones (iPhone 7 plus) under real conditions from the planting base of the Pomology Institute of Shanxi Agricultural University and farmers’ orchards, and the resolution of the images were 3024 × 4032. The disease images were captured between May and September 2020. The original dataset was prepared with 2141 images, which included 597 images of healthy leaves, 592 images of scab, 622 images of rust, 153 images of ring rot, and 177 images of leaves with panonychus ulmi. These images were captured with uneven illumination intensities and heterogeneous field in wild scenarios. All the images collected in this paper are defined according to their disease categories and have been saved in the JPEG format. Each image contains only one apple leaf, and the leaf occupies the main position in the picture. We chose to cooperate with plant protection experts to establish a more reliable apple leaf disease dataset. Five major diseases due to pests and pathogens include rust, scab, ring rot, and panonychus ulmi; these are shown in Figure 1, along with a healthy sample. The characteristic symptoms of the above-mentioned apple leaf diseases are given as follows.

Rust: In the early stage of the disease, small orange-red spots appear on the front of the leaves, gradually expanding to form an orange-yellow spot with red edges. When the disease is serious, dozens of disease spots appear on one leaf. The surface of the spot becomes dense with small bright yellow dots 1~2 weeks after the onset of disease, indicating sexual sporozoite.

Scab: In the early stage of the disease, spots are yellow-green round or radial, then turn brown to black, with obvious edges. When the disease is serious, the leaf becomes smaller and thicker, curled or twisted. Disease spots will merge with one another and leafs will show big spots.

Apple ring rot: Lesions are concentric whorls which are mostly concentrated in the leaf margin. In the early stage of the disease, the spot is brown to dark brown, round, and small; as the disease advances, the spots enlarge. The spots on the leaf margin are semicircular, and on the middle of the leaf are round or nearly round. The central part of the old disease spot is gray-brown to gray-white, and the disease spot is often broken or perforated.

Panonychus ulmi: The affected leaf is covered with yellow-white spots on the front, and ultimately, the whole leaf withers and yellows.

#### 2.1.2. Image Pre-Processing

Image pre-processing helps to enhance the quality of data to promote the extraction of meaningful insights from the data. It includes a redefinition of image size, normalization of each batch of samples, normalization processing, image de-averaging, data augmentation, and automatic denoising [26,27].

The normalization processing method for a sample is shown in Equations (1) and (2):(1)$$\mu =\frac{1}{n}\sum_{i=1}^{n}x_{i}$$
(2)$$\sigma^{2}=\frac{1}{n}\sum_{i=1}^{n}(x_{i}-\mu {)}^{2}$$
where $x_{i}$ represents the value of the *i*-th pixel of the sample, *n* is the total number of pixels of the sample, μ represents the mean value and $\sigma^{2}$ represents the variance. The normalization formula is shown in Equation (3)
(3)$$\overset{\wedge }{x_{i}}=\frac{x_{i}-\mu }{\sqrt{\sigma^{2}+\varepsilon }}$$
where $\overset{\wedge }{x_{i}}$ is the normalization of the *i*-th pixel value of the sample, and ε is a tiny value greater than 0 to ensure that the denominator is greater than 0.

In the normalization processing image de-averaging, the R, G and B values of the images were respectively subtracted from their mean values R_MEAN, G_MEAN B_MEAN, where R_MEAN is the pixel mean of the R channel of all images, G_MEAN is the pixel mean of the G channel of all images, and B_MEAN is the pixel mean of the B channel of all images.

Due to the fact that the apple leaf disease dataset collected is not balanced among classes, data enhancement techniques, such as image rotation, horizontal translation, vertical translation, brightness adjustment, and image scaling, were used to further expand the image data set and finally obtain a balanced data set. The balanced data set was randomly divided into a training set, validation set, and test set according to the ratio of 7:2:1.

Although deep learning neural networks are very powerful, if there are not enough images, overfitting will occur, which cannot achieve the desired results [28]. Many researchers have done a lot of work on this topic. Therefore, in order to avoid overfitting, data augment of the training set and validation set is carried out. The data distributions of the original dataset, balanced dataset, training set, validation set, and test set are shown in Table 1.

### 2.2. Lightweight Convolutional Neural Network Models

In this part, we describe in detail the lightweight convolutional neural network model, RegNet, used in our work.

#### 2.2.1. RegNet

In recent years, NAS [29] (neural architecture search) network search technology has been very popular, but it also has high requirements for computing resources. The traditional NAS method, which is based on individual network instances (sampling one network at a time), has the following defects: (1) it is very inflexible (various parameter adjustment methods), (2) it has a poor ability to generalize, and (3) it has poor interpretability. Therefore, scholars have put forward the overall estimation of network design space (population estimation, which means that the optimal design space relationship, such as depth and width, is estimated). Intuitively, if we can identify the functional relationship between a series of network design elements such as depth and width on the network design goal, then we can easily know how deep and wide of a network is the best choice. The design idea of EfficientNet is to use a simple and efficient composite coefficient to enlarge the network from the three dimensions of depth, width, and resolution. This method will not arbitrarily scale the dimensions of the network like the traditional method. NAS can obtain the best set of parameters (composite coefficient) based on the neural structure search technology, and explore how to search for the best model under a certain computational cost.

RegNet also uses NAS technology, but differently from some previous NAS (such as MobileNetV3, EfficientNet). Some of the previous NAS used search algorithms to find the best combination of parameters in a given designed search space. However, what RegNet aims to explore is how to design spaces and find some network design principles, rather than just searching for a set of parameters. RegNet is not a single network, nor even an extended family of networks like the EfficientNets. It is a design space limited by quantized linear rules. With the same training design and FLOPs, RegNet is more accurate than the current state-of-the-art (SOTA) EfficientNet and is five times faster on GPU than EfficientNet. The structural framework of the RegNet network is shown in Figure 2.

Figure 2a shows that the RegNet network is mainly composed of three parts: stem, body, and head. This network keeps the backbone and head network as simple as possible, mainly focusing on the structure of the network body.

The stem is an ordinary convolution layer (including BN and Relu by default). The size of the convolution kernel is 3 × 3, the step length is 2, and the number of convolution cores is 32.

The structure of the body, shown in Figure 2b, is made up of four stages, like a stack. After each stage, as shown in Figure 2c, the height and width of the input characteristic matrix will be reduced to half of the original. Each stage is composed of a series of block stacks. In the first block of each stage, there are group convolutions (on the main branch) and ordinary convolutions (on the shortcut branch) with a step of 2, and the step of convolutions in the remaining blocks are 1.

Head is a common classifier in the classification network and is composed of a global average pooling layer and a full connection layer.

Figure 3 shows the structure of the block, in which Figure 3a shows the case of step stripe = 1 and Figure 3b shows the case of step stripe = 2. As can be seen from Figure 3, the block in RegNet is basically the same as the block in ResNext. The main branches are a 1 × 1 convolution (including BN and ReLU), a 3 × 3 group convolution (including BN and ReLU), and a 1 × 1 convolution (including BN). On the shortcut branch, when stripe = 1, no processing is performed. When stripe = 2, downsampling, is performed through a 1 × 1 convolution (including BN). The r in Figure 3 represents the resolution, which can be simply understood as the height and width of the characteristic matrix. When the step distance s equals 1, the input and output r remains unchanged; when s equals 2, the output r is half of the input. W is the channel of the characteristic matrix (note that when s = 2, the input is W_i-1_ and the output is W_i_; that is, the channel will change), g represents the group width of each group in the group convolution, and b represents the bottleneck ratio; that is, the channel of the output characteristic matrix is reduced to 1/b of the input characteristic matrix channel.

#### 2.2.2. Ranger

The current state-of-the-art optimizer Ranger was proposed by Dang [30]. The Ranger optimizer combines the two emerging achievements of RAdam [31] and Lookahead [32] to build a set of optimizers for deep learning. Among them, Radam (Rectified Adam), a variant of Adam, can be said to be a better foundation for the optimizer at the beginning of training. RAdam uses a dynamic rectifier to adjust Adam’s adaptive momentum based on variance and effectively provides an automatic warm-up mechanism that can be customized based on the current data set, which can ensure that training takes a solid first step. LookAhead can provide a robust and stable breakthrough throughout training. Additionally, Lookahead can reduce the need for hyperparametric fine-tuning and achieve faster convergence between different deep learning tasks. Therefore, higher accuracy can be obtained by combining the two.

### 2.3. Transfer Learning

Transfer learning is a machine learning technique where knowledge gained during training in one type of problem is used to train other related tasks or domains [33]. First, the structure of the convolution layers is kept unchanged, and then the pre-trained weights and parameters are loaded into the convolution layers. Then, a new fully-connected layer is designed for the new task, which replaces the original fully-connected layer and forms a new convolution network model with the previous convolution layers. This model is then fine-tuned on the target data set. There are two ways to fine-tune; one is freezing the convolutional layers and fine-tuning the fully connected layers only, while the other is to fine-tune the entire layers of the network.

The training of a deep convolutional neural network requires a huge dataset. Considering that the apple leaf disease dataset is not huge enough, the pre-trained RegNet, MobileNetV3, ShuffleNet, and EfficientNet-B0 models are applied to the classification of apple leaf disease images with complex backgrounds by using the transfer learning method.

### 2.4. Experiment Setup

The experiments were conducted in the environment of Ubuntu18.04 with an Intel Core i9 9820X, 64 G memory, and a GeForce RTX 2080Ti 11 G DDR6 using the deep learning framework TensorFlow and Cuda10.1 for training.

In order to avoid memory overflow, a batch training method was adopted to compare the RegNet model with EfficientNet-B0, MobileNetV3, and ShuffleNet on the training set and validation set. Each batch was trained with 16 images, and the batch size of the validation set was also 16 images. In this paper, the number of iterations was set to 50, and the categorical-cross entropy in Keras was used as the cost function. Batch normalization was introduced to standardize the input of the hidden layer.

In order to save the optimal model parameters, after each iteration, whether the loss function of the validation set was reduced or not was observed to decide whether to save the current model parameters. Finally, the saved model structure and parameters were used for the prediction of apple leaf disease on the test set.

## 3. Experiments and Analysis

However, we note that the model performance may be affected not only by the architecture choice but also by other parameters, such as the training schedule, optimizers, and learning rate. We perform three series of experiments to evaluate the learning capabilities of ShuffleNet, MobileNetV3, EfficientNet-B0, and RegNet with different transfer learning methods, different optimizers, and different learning rates. To keep the parameters of the RegNet model at the same level as MobileNetV3 and EfficientNet-B0, in this section we used the RegNet model with a 400 million flops (400 MF) regime and pre-trained each sampled model for 10 epochs on the ImageNet dataset.

### 3.1. Comparison of Two Transfer Learning Methods

Two transfer learning strategies were considered as follows. In the first one, the weights of different deep learning network models were initialized by using the model parameter files which were pre-trained on the ImageNet dataset instead of the original random initialization operation, and the global fine-tuning was carried out. The other utilized pre-trained neural network architectures such as ShuffleNet, MobileNetV3, EfficientNet-B0, and RegNet as feature extractors and fine-tuned the higher dimensional layers (fully connected layers) to learn features corresponding to the target dataset. For all the models, the learning rate was set to 0.0001 and the optimizer was Adam.

Figure 4 summarizes the average accuracy and loss function value using different kinds of transfer learning methods on the validation set. In Figure 4, the solid line represents the use of transfer learning strategy one, and the dotted line represents the use of transfer learning strategy two. Several interesting observations can be drawn from the results. From the contrast in Figure 4, we can draw a conclusion that for all models, fine-tuning the parameters of a pre-trained neural network architecture achieved higher classification accuracy and lower loss values as compared to using the neural network architecture with feature extraction only. The reason is that the image features can be extracted by the convolution layer module with the weight of the transfer parameters, but there are great differences between the apple leaf disease images in this study and the ImageNet image data. Only training and changing the full connection module cannot achieve the ideal effect, and transfer learning training all layers can significantly improve the accuracy of the test set. This showed that the training classification layers alone cannot make the model adapt to our data well.

Therefore, it is necessary to adopt the transfer learning method that fine-tunes the whole layers on our apple leaf disease image dataset.

### 3.2. Evaluation of the Optimizers

Convenient deep learning architectures such as ShuffleNet, MobileNet, EfficientNet, and RegNet are typically trained with SGD, and we tried using other optimizers, such as Adam, RAdam, and Ranger. Namely, we compared the fine-tuning performance of the four models pre-trained on ImageNet with SGD, Adam, Radam, and Ranger. For all the models, we used a learning rate of 0.0001 and used transfer learning of fine-tuning on the whole models. The results are presented in Table 2, where the average accuracy is the average accuracy on the validation set.

As can be seen from Table 2, from the perspective of the model, ShuffleNet had the smallest number of parameters and the lowest classification accuracy compared with the other three models, and the performance of the four optimizers was quite different from each other. This showed that the model was sensitive to the choice of the optimizers. The best optimizer is the Adam optimizer, which had a recognition accuracy of 90.2%, and the SGD optimizer had the lowest recognition accuracy of 62.3%. The other two optimizers, RAdam and Ranger, also had much lower identification accuracy than the Adam optimizer. For MobileNetV3, EfficientNet-B0, and RegNet, the recognition accuracy of the optimizer SGD was also the worst, while the other three optimizers, Adam, RAdam, and Ranger, were almost at the same level. This showed that the model was insensitive to the choice of these three optimizers.

The above analysis showed that the model trained by the SGD optimizer had the worst performance; it may be that the SGD optimization algorithm adjusts the weights for each data point, and the performance of the network fluctuates significantly during the training process, which was consistent with the research results in the literature. Therefore, the SGD optimizer will not be considered later.

The accuracy curves of the Adam, RAdam, and Ranger optimizers on the validation set for the four models are shown in Figure 5a–c, respectively.

According to Figure 5, from the sensitivity of the models to the optimizers, the four models trained by the Adam optimizer all showed good performance. The average accuracy of the RegNet model trained by the Adam optimizer is as high as 99.8%. When Figure 5b,c are compared with Figure 5a, it can be seen that the convergence rate of the four models is slower when using the Radam and Ranger optimizers than when using Adam, and there is a great difference between the best accuracy of the models. When the Adam optimizer is used, the four models achieve the best convergence state, and the differences between the best accuracy of the models are small. This showed that the models trained by the Adam optimizer can obtain a better classification performance; that is, the optimizer Adam had better universality to the models.

The analysis of these results concluded that RegNet outperformed all other lightweight pre-trained models under the three optimizers, and ShuffleNet performed worst. Therefore, the RegNet model, with its good performance, and the Adam optimizer, with its good universality, will be selected for follow-up studies.

### 3.3. Effect of Learning Rate on Model Performance

The learning rate is a very important hyperparameter in the training process of a convolutional neural network. It represents the range of parameters updated each time, and a suitable learning rate can accelerate the convergence speed of model training. The RegNet model with the Adam optimizer was used to select the most appropriate learning rate for the apple leaf disease dataset among 6 groups of learning rates of 0.0001, 0.0005, 0.001, 0.005, 0.01, and 0.05.

Figure 6a,b shows the accuracy and loss function curves of the model on the validation set under different learning rates. The accuracy of the model on the validation set and test set under different learning rates is shown in Table 3.

It can be seen from Figure 6 that when the learning rates were 0.05, 0.01 and 0.005, the curve oscillated seriously, converged slowly, and the loss values of the model were higher and the accuracies were lower. When the learning rate was 0.05, the curve oscillated most severely, and the model had not converged when it reached the maximum number of iterations, indicating that the learning rate was too high; the parameters that need to be optimized will fluctuate without convergence. According to Table 3, the accuracy of the model was as low as 94.61% when the learning rate was 0.05.

When the learning rates were 0.001, 0.0005, and 0.0001, the oscillation of the curve was not obvious, and each model could obtain higher accuracy and lower loss value. When the learning rate was 0.0001, the loss value curve and average accuracy curve of the model are more stable, with lower loss value and higher accuracy than those of the model when the learning rates were 0.001 and 0.0005. According to Table 3, when the learning rate was 0.0001, the model achieved the highest recognition accuracy of 99.23%.

In summary, in the fine-tuning training stage of the model, when the initial learning rate was set to a small value, the performance of the training model was better. The reason was that under the transfer learning mode, all layers at the front-end layer of the network had been well trained, and the weight parameters of the model were close to the optimal solution. If a large learning rate was used in the fine-tuning training stage, it was easy to cause the model to skip the optimal solution and produce a large oscillation, so as to increase the loss function value and reduce the accuracy value. As the RegNet-Adam-0.0001 model had the highest test accuracy of 99.23% and the lowest loss value on the test set of apple leaf disease, this showed that the performance of the model was the best.

To further observe the performance of the improved model on the test set, the confusion matrix of RegNet-Adam-0.0001 on the test set is shown in Figure 7.

Each column in Figure 7 represents the predicted labels, and its total number indicates the number of samples predicted for that category. Each row represents the real labels, and the total number indicates the number of samples of the true data belonging to the category. The value at the intersection of the row and column represents the number of data predicted as the corresponding row category, and the sum of the values in the diagonal line is the predicted correct results. The deeper the color in the visualization results, the higher the prediction accuracy of the model in the corresponding class. Based on the confusion matrix, three metrics, precision, recall, and specificity, were used to evaluate the performances of the methods, and the results were shown in Table 4.

According to the confusion matrix in Figure 7, the predicted categories are consistent with the true categories of these images, and most apple leaf diseases are correctly classified by the proposed approach with a high probability except for two images. That is, 48 sample images are correctly identified in the class of “Healthy”, except for 1 misclassified sample. For the classification of “scab” in 46 samples, 45 were correctly classified, and the other three categories were correctly classified. Thus, there are a total of 258 apple leaf disease images that are correctly classified by the proposed approach in 260 samples.

From the results in Table 4, we can see that the precision values were between 0.98 and 1.0, with an average value of 0.9924. The recall values were between 0.978 and 1.0, with an average of 0.9916. The specificity values were between 0.995 and 1.0 with an average of 0.9980, and the average accuracy of the model was 99.23%. This indicates that the proposed RegNet-Adam-0.0001 approach had a significant capability to recognize the apple leaf diseases.

### 3.4. Comparison with State-of-the-Art CNNs

In this section, we will compare RegNet with SOTA CNNs (Vision Transformer) on our apple leaf disease dataset.

Inspired by the Transformer scaling successes in NLP, the Google team published a paper in October 2020: “An Image is Worth 16 × 16 Words: Transformers for Image Recognition at Scale”, and the ViT was first proposed in the paper. ViT applies a standard transformer directly to images with the fewest possible modifications. To do so, they split an image into patches and provide the sequence of linear embeddings of these patches as an input to a transformer. As ViT has nothing to do with the structure of input elements, the authors added a learnable location embedding in each patch to enable the model to understand the image structure. Therefore, image patches are treated the same way as tokens (words) in an NLP application. ViT attained excellent results compared to state-of-the-art CNNs on ImageNet, CIFAR-100, and VTAB.

Due to the large number of parameters and calculations of the existing ViT model, it is difficult to train it directly. Therefore, the transfer learning method of fine-tuning all layers on our apple leaf disease images dataset was used. For all models, the Adam optimizer was used, and the learning rate was set to 0.0001.

The average accuracy and loss function curves of the models ViT-B/16 and RegNet on the training set and validation set were shown in Figure 8. The black curve was the performance curve of the RegNet model, and the red curve and the blue curve were the performance curves of ViT model on the training set and validation set, respectively. ViT-B/16 means the “Base” variant with a 12 × 12 input patch size.

By comparing the accuracy curves of the two models in Figure 8 on the validation set, it can be seen that the two models have reached a stable convergence state at the maximum iterative training, indicating that the models had been fully trained. The verification accuracy of RegNet and ViT-B/16 were 99.8% and 93.3%, respectively, and the verification losses were 0.023 and 0.33, respectively. The RegNet model had the highest accuracy and the lowest loss value, which indicated that the performance is best. By comparing the accuracy curves of the ViT-B/16 model on the training set and the validation set in Figure 8a, we find that the accuracy on the training set was higher than the validation set, indicating that there is overfitting. There may be two reasons for this phenomenon. On the one hand, the ViT model still directly applies the transformer structure in natural language processing (NLP), there is no special design for the image, and there is not enough image-related knowledge built in. On the other hand, the information of an image is much greater than that of the text, resulting in a large number of parameters and calculations of the ViT model. The parameters of the ViT-B/16 model are close to four times those of the RegNet model, and our data set is relatively small, so we cannot train the model enough, resulting in slow convergence and serious overfitting. Therefore, it is necessary to further optimize the ViT model in the future to make it better serve the field of computer vision.

## 4. Discussion

In order to realize the rapid and accurate identification of apple leaf disease, a new lightweight convolutional neural network, RegNet, was proposed to identify five apple leaf diseases (rust, scab, ring rot, panonychus ulmi, and healthy leaves) in the field complex environment. At present, some scholars have applied convolutional neural networks to recognize apple leaf disease images and achieved good results [34,35,36]. However, there are some shortcomings. For example, Liu et al. [34] detected apple leaf diseases with simple backgrounds. It is difficult generalize this ability to complex backgrounds. The shortcoming is that only two kinds of apple leaf diseases were studied [35]. The deep learning models used by Jiang et al. [36] are large and require large memory, which is difficult to deploy on embedded or mobile devices with limited hardware resources.

Compared with the research on apple leaf diseases mentioned above, the object of our work was to detect five kinds of apple leaf diseases with complicated backgrounds in the field, and the test accuracy was as high as 99.23%. In addition, the RegNet used was a lightweight network model that can be well embedded into mobile devices. The RegNet model still had the highest accuracy and the lowest loss value compared with the state-of-the-art ViT-B/16 model.

Overall, our model had good general performance and high identification accuracy for apple leaf diseases. Since this model is a lightweight network model, it can realize apple leaf disease detection in the mobile phone terminal. Thus, this work has practical application value.

## 5. Conclusions

In this work, the lightweight deep learning model RegNet was developed and compared to different lightweight CNN architectures such as MobileNetV3, ShuffleNet, ViT-B/16, and EfficientNet-B0 for the efficient classification of apple leaf diseases based on collected apple leaf images of healthy leaves and four categories of diseases with complex backgrounds. The effects of different training mechanisms, learning rates, and optimizers on the performance of the model were discussed. The conclusions are as follows:Fine-tuning the parameters of a pre-trained neural network architecture achieved higher classification accuracy as compared to using the neural network architecture with feature extraction only;We compared the performance of the four models with SGD, Adam, RAdam, and Ranger. This comparison showed that the models trained by the Adam optimizer can obtain a better classification performance, and the average accuracy of the RegNet model trained by the Adam optimizer reached as high as 99.8% on the validation set;Comparative experiments of different learning rates were carried out on RegNet-Adam, and the results showed that when the initial learning rate was set to 0.0001, and the effect is better than 0.05, 0.005, 0.0005, 0.001, and 0.01, the accuracy reached 99.23% on the test set. Comparing RegNet-Adam-0.0001 with the ViT-B/16 model led to the same conclusion that the RegNet model had the best performance.

Thus, the proposed approach efficiently accomplished apple leaf disease identification and presented a superior performance relative to other SOTA methods. In future developments, we intend to deploy this model on mobile devices to monitor and identify the wide range of apple leaf disease information automatically.

## Figures and Tables

**Figure 1 sensors-22-00173-f001:**
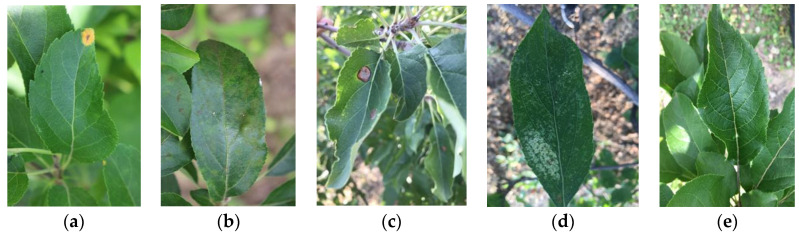
Example of apple leaf images: (**a**) rust; (**b**) scab; (**c**) ring rot; (**d**) Panonychus ulmi; (**e**) healthy.

**Figure 2 sensors-22-00173-f002:**
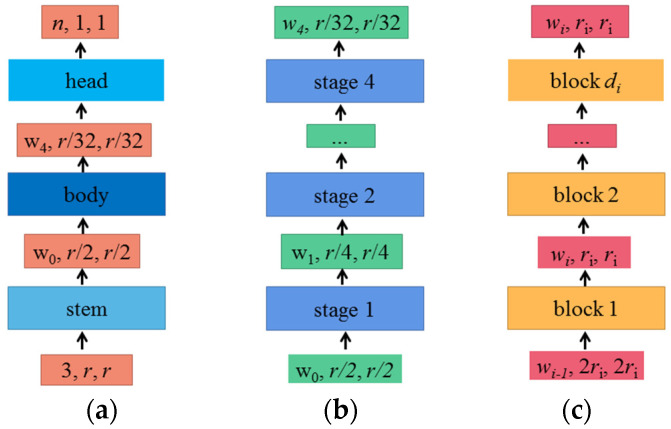
Structural framework of RegNet: (**a**) network; (**b**) body; (**c**) stage.

**Figure 3 sensors-22-00173-f003:**
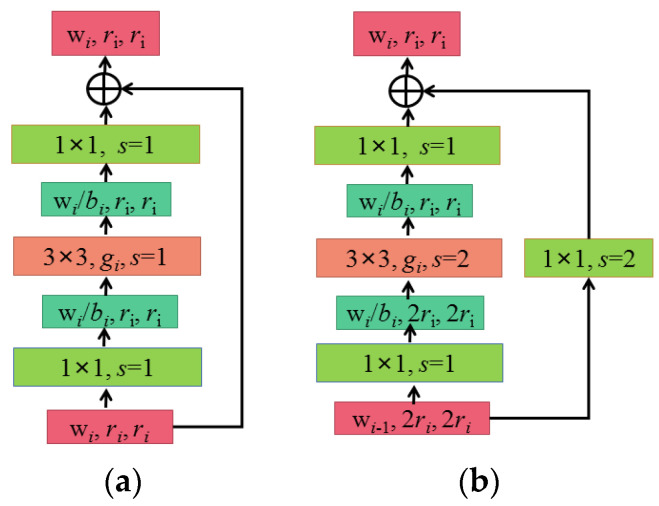
Structural framework of the block: (**a**) X block, s = 1; (**b**) X block, s = 2.

**Figure 4 sensors-22-00173-f004:**
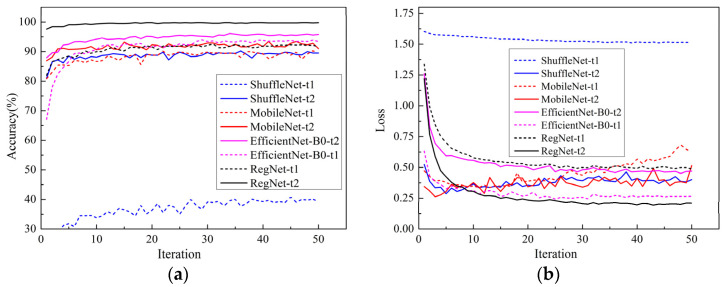
Comparison of two transfer learning methods: (**a**) the average accuracy curve; (**b**) the loss function curve.

**Figure 5 sensors-22-00173-f005:**
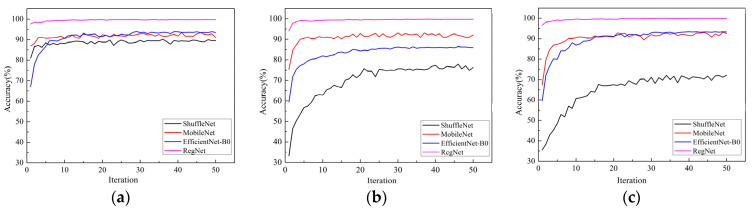
Comparison of different models under the same optimizer: (**a**) Adam; (**b**) RAdam; (**c**) Ranger.

**Figure 6 sensors-22-00173-f006:**
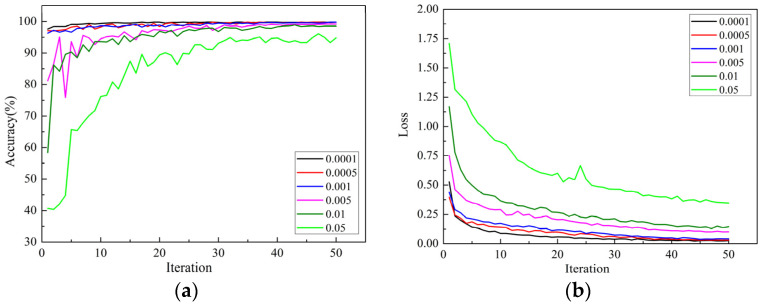
The influence of learning rate on the model training effect: (**a**) average accuracy curve of the models; (**b**) loss curve of the models.

**Figure 7 sensors-22-00173-f007:**
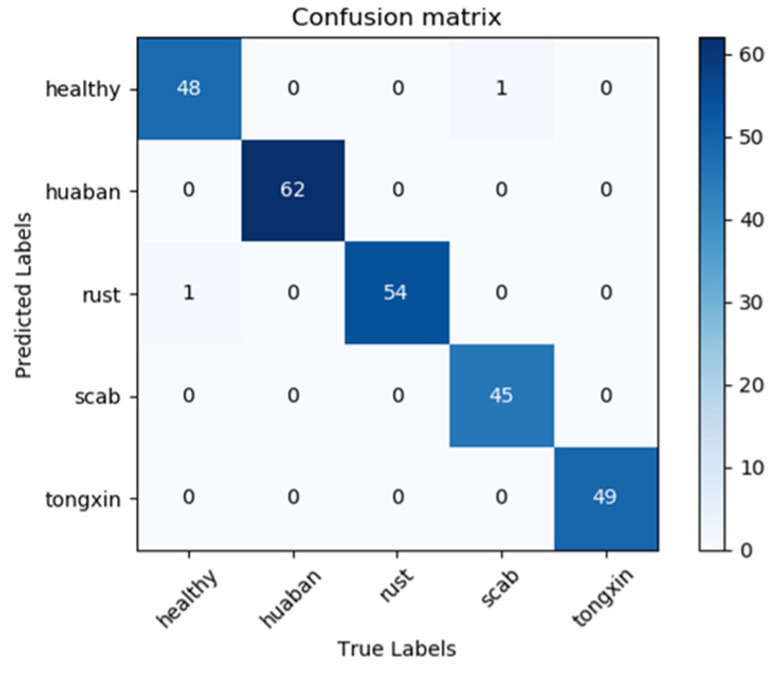
The confusion matrix of the test set.

**Figure 8 sensors-22-00173-f008:**
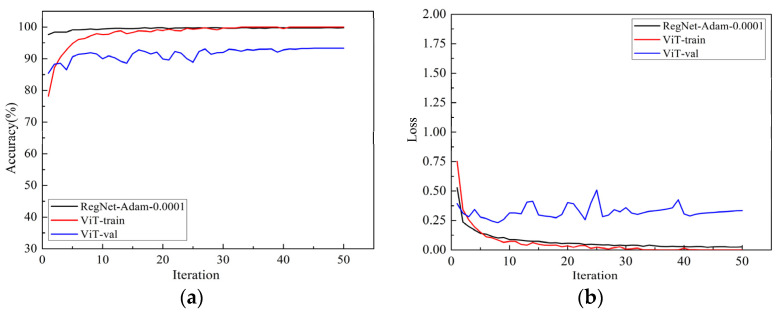
The performance of the two models: (**a**) the accuracy curve of the models; (**b**) the loss curve of the models.

**Table 1 sensors-22-00173-t001:** The data distributions.

Classes	Original Dataset	Balanced Dataset	Training Set	Validation Set	Augmented Dataset	Test Set
Healthy	597	597	429	119	1644	49
Scab	592	592	428	118	1638	46
Rust	622	622	444	124	1704	54
Ring rot	153	612	441	122	1641	49
Panonychus ulmi	177	708	505	141	1883	62
Total	2141	3131	2247	624	8510	260

**Table 2 sensors-22-00173-t002:** Performance comparison of different optimizers for four network models.

Model	Optimizers	Params	Average Accuracy (%)
ShuffleNet	SGD	2.3 M	62.3
	Adam		90.2
	RAdam		77.9
	Ranger		72.1
MobileNetV3	SGD	5.4 M	89.9
	Adam		93.4
	RAdam		93.1
	Ranger		93.2
EfficientNet-B0	SGD	5.3 M	86.4
	Adam		94.3
	RAdam		94
	Ranger		93.4
RegNet	SGD	5.2 M	98.2
	Adam		99.8
	RAdam		99.8
	Ranger		99.9

**Table 3 sensors-22-00173-t003:** The average accuracy of the model with different learning rates.

Pre-Trained Models	Accuracy on Validation Set (%)	Accuracy on Test Set (%)
RegNet-Adam-0.0001	99.8	99.23
RegNet-Adam-0.0005	99.8	99.20
RegNet-Adam-0.001	99.8	98.07
RegNet-Adam-0.005	99.2	98.84
RegNet-Adam-0.01	98.9	98.07
RegNet-Adam-0.05	96.1	94.61

**Table 4 sensors-22-00173-t004:** The results of different metrics.

Category	Precision	Recall	Specificity
Rust	0.982	1.000	0.995
Scab	1.000	0.978	1.000
Ring rot	1.000	1.000	1.000
Panonychus ulmi	1.000	1.000	1.000
Healthy	0.980	0.980	0.995
Average value	0.9924	0.9916	0.9980

## Data Availability

Not applicable.

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
