# Peer review of "Apple Leaf Disease Identification with a Small and Imbalanced Dataset Based on Lightweight Convolutional Networks"

_sensors, 2021, doi:10.3390/s22010173_

Round 1

Reviewer 1 Report

The paper deals with the use of IA-based methods for identification of apple leaf diseases.

The paper aims to overcome the problem of possible misjudges in identification and classification, which leads to massive use of pesticides, thus affecting the quality of the product.

The paper is well written and scientifically sound. Authors introduces the problem, analyse the soa and provide the method basically consisting on testing and fine tuning a number of CNNs. Moreover they compare the performance of their method with soa CNNs.

I have some minor comments:

1) Abstract should be shortened since it exceeds the limit of 200 words consistently.

2) In Section 2.1.1. which is the hardware used for acquiring the dataset? In Section 2.1.2 some methods to enhance the quality of data are mentioned. However, no description of which method is used is provided. More in particular authors should mention the resolution of the images used as a dataset, the numerof data… more details.

3) Maybe due to PDF conversion, the quality of figures should be strongly enhanced.

4) Authors could consider to mention in the literature review also similar studies on olives (e.g. ANN-based method for olive Ripening Index automatic prediction (2010) Journal of Food Engineering, 101 (3), pp. 318-328) – not mandatory.

Reviewer 2 Report

===== Synopsis:

Authors apply a RegNet model to identify 4 classes of leaf diseases (in apple trees). They thoroughly evaluate the model and make a comparison to the latest networks. The results are similar to the vision transformer network but the authors' system is more efficient.

===== General Comments:

The study reads well. I have only few comments.

The entire discussion section 4 belongs to the introduction in order to motivate the use of the RegNet, as the last sentence of the section expresses it.

The actual discussion starts at line 429 already with "There may be two reasons for this''.

===== Specific Comments:

Line 151: NAS. Lacks citation. It took me a while to understand what it is about.

Figures 4 & 5 & 6 are hardly readable: I suspect the pdf conversion process degraded the quality; or authors include a jpeg file and not an eps file. For publication this must be higher resolution.

Lines: 320-2: while the performance...sentence is akward. It appears written in haste.

Lines 451: "The shortcoming is that only two kinds of apple leaf diseases were studied, which is a simple binary classification problem". Not really. binary would be healthy/affected discrimination. With two disease classes, it becomes a three-class discrimination task.
